# Thinking Fast and Slow
# with Deep Learning and Tree Search

**Thomas Anthony**[1, ✉], **Zheng Tian**[1], and **David Barber**[1,2]

[1]University College London
[2]Alan Turing Institute
✉thomas.anthony.14@ucl.ac.uk

## Abstract

Sequential decision making problems, such as structured prediction, robotic control, and game playing, require a combination of planning policies and generalisation of those plans. In this paper, we present Expert Iteration (ExIt), a novel reinforcement learning algorithm which decomposes the problem into separate planning and generalisation tasks. Planning new policies is performed by tree search, while a deep neural network generalises those plans. Subsequently, tree search is improved by using the neural network policy to guide search, increasing the strength of new plans. In contrast, standard deep Reinforcement Learning algorithms rely on a neural network not only to generalise plans, but to discover them too. We show that ExIt outperforms REINFORCE for training a neural network to play the board game Hex, and our final tree search agent, trained tabula rasa, defeats MoHex 1.0, the most recent Olympiad Champion player to be publicly released.

## 1 Introduction

According to dual-process theory [1, 2], human reasoning consists of two different kinds of thinking. *System 1* is a fast, unconscious and automatic mode of thought, also known as *intuition* or *heuristic process*. *System 2*, an evolutionarily recent process unique to humans, is a slow, conscious, explicit and rule-based mode of *reasoning*.

When learning to complete a challenging planning task, such as playing a board game, humans exploit both processes: strong intuitions allow for more effective analytic reasoning by rapidly selecting interesting lines of play for consideration. Repeated deep study gradually improves intuitions. Stronger intuitions feedback to stronger analysis, creating a closed learning loop. In other words, humans learn by *thinking fast and slow*.

In deep Reinforcement Learning (RL) algorithms such as REINFORCE [3] and DQN [4], neural networks make action selections with no lookahead; this is analogous to System 1. Unlike human intuition, their training does not benefit from a 'System 2' to suggest strong policies. In this paper, we present Expert Iteration (ExIt), which uses a Tree Search as an analogue of System 2; this assists the training of the neural network. In turn, the neural network is used to improve the performance of the tree search by providing fast 'intuitions' to guide search.

At a low level, ExIt can be viewed as an extension of Imitation Learning (IL) methods to domains where the best known experts are unable to achieve satisfactory performance. In IL an *apprentice* is trained to imitate the behaviour of an *expert* policy. Within ExIt, we iteratively re-solve the IL problem. Between each iteration, we perform an expert improvement step, where we bootstrap the (fast) apprentice policy to increase the performance of the (comparatively slow) expert.

Typically, the apprentice is implemented as a deep neural network, and the expert by a tree search algorithm. Expert improvement can be achieved either by using the apprentice as an initial bias in the search direction, or to assist in quickly estimating the value of states encountered in the search tree, or both.

We proceed as follows: in section 2, we cover some preliminaries. Section 3 describes the general form of the Expert Iteration algorithm, and discusses the roles performed by expert and apprentice.

Sections 4 and 5 dive into the implementation details of the Imitation Learning and expert improvement steps of EXIT for the board game Hex. The performance of the resultant EXIT algorithm is reported in section 6. Sections 7 and 8 discuss our findings and relate the algorithm to previous works.

## 2   Preliminaries

### 2.1   Markov Decision Processes

We consider sequential decision making in a Markov Decision Process (MDP). At each timestep $t$, an agent observes a state $s_t$ and chooses an action $a_t$ to take. In a terminal state $s_T$, an episodic reward $R$ is observed, which we intend to maximise.[1] We can easily extend to two-player, perfect information, zero-sum games by learning policies for both players simultaneously, which aim to maximise the reward for the respective player.

We call a distribution over the actions $a$ available in state $s$ a *policy*, and denote it $\pi(a|s)$. The value function $V^\pi(s)$ is the mean reward from following $\pi$ starting in state $s$. By $Q^\pi(s, a)$ we mean the expected reward from taking action $a$ in state $s$, and following policy $\pi$ thereafter.

### 2.2   Imitation Learning

In Imitation Learning (IL), we attempt to solve the MDP by mimicking an *expert* policy $\pi^*$ that has been provided. Experts can arise from observing humans completing a task, or, in the context of structured prediction, calculated from labelled training data. The policy we learn through this mimicry is referred to as the *apprentice* policy.

We create a dataset of states of expert play, along with some target data drawn from the expert, which we attempt to predict. Several choices of target data have been used. The simplest approach is to ask the expert to name an optimal move $\pi^*(a|s)$ [5]. Once we can predict expert moves, we can take the action we think the expert would have most probably taken. Another approach is to estimate the action-value function $Q^{\pi^*}(s, a)$. We can then predict that function, and act greedily with respect to it. In contrast to direct action prediction, this target is cost-sensitive, meaning the apprentice can trade-off prediction errors against how costly they are [6].

## 3   Expert iteration

Compared to IL techniques, Expert Iteration (EXIT) is enriched by an expert improvement step. Improving the expert player and then resolving the Imitation Learning problem allows us to exploit the fast convergence properties of Imitation Learning even in contexts where no strong player was originally known, including when learning tabula rasa. Previously, to solve such problems, researchers have fallen back on RL algorithms that often suffer from slow convergence, and high variance, and can struggle with local minima.

At each iteration $i$, the algorithm proceeds as follows: we create a set $\mathcal{S}_i$ of game states by self play of the apprentice $\hat{\pi}_{i-1}$. In each of these states, we use our expert to calculate an Imitation Learning target at $s$ (e.g. the expert's action $\pi^*_{i-1}(a|s)$); the state-target pairs (e.g. $(s, \pi^*_{i-1}(a|s))$) form our dataset $\mathcal{D}_i$. We train a new apprentice $\hat{\pi}_i$ on $\mathcal{D}_i$ (Imitation Learning). Then, we use our new apprentice to update our expert $\pi^*_i = \pi^*(a|s; \hat{\pi}_i)$ (expert improvement). See Algorithm 1 for pseudo-code.

The expert policy is calculated using a tree search algorithm. By using the apprentice policy to direct search effort towards promising moves, or by evaluating states encountered during search more quickly and accurately, we can help the expert find stronger policies. In other words, we bootstrap the knowledge acquired by Imitation Learning back into the planning algorithm.

The Imitation Learning step is analogous to a human improving their intuition for the task by studying example problems, while the expert improvement step is analogous to a human using their improved intuition to guide future analysis.

---

**Algorithm 1** Expert Iteration

1: $\hat{\pi}_0$ = initial_policy()
2: $\pi_0^*$ = build_expert($\hat{\pi}_0$)
3: **for** i = 1; i $\leq$ max_iterations; i++ **do**
4:     $S_i$ = sample_self_play($\hat{\pi}_{i-1}$)
5:     $D_i = \{(s, \text{imitation\_learning\_target}(\pi_{i-1}^*(s))) | s \in S_i\}$
6:     $\hat{\pi}_i$ = train_policy($D_i$)
7:     $\pi_i^*$ = build_expert($\hat{\pi}_i$)
8: **end for**

---

## 3.1   Choice of expert and apprentice

The learning rate of EXIT is controlled by two factors: the size of the performance gap between the apprentice policy and the improved expert, and how close the performance of the new apprentice is to the performance of the expert it learns from. The former induces an upper bound on the new apprentice's performance at each iteration, while the latter describes how closely we approach that upper bound. The choice of both expert and apprentice can have a significant impact on both these factors, so must be considered together.

The role of the expert is to perform exploration, and thereby to accurately determine strong move sequences, from a single position. The role of the apprentice is to generalise the policies that the expert discovers across the whole state space, and to provide rapid access to that strong policy for bootstrapping in future searches.

The canonical choice of expert is a tree search algorithm. Search considers the exact dynamics of the game tree local to the state under consideration. This is analogous to the lookahead human games players engage in when planning their moves. The apprentice policy can be used to bias search towards promising moves, aid node evaluation, or both. By employing search, we can find strong move sequences potentially far away from the apprentice policy, accelerating learning in complex scenarios. Possible tree search algorithms include Monte Carlo Tree Search [7], $\alpha$-$\beta$ Search, and greedy search [6].

The canonical apprentice is a deep neural network parametrisation of the policy. Such deep networks are known to be able to efficiently generalise across large state spaces, and they can be evaluated rapidly on a GPU. The precise parametrisation of the apprentice should also be informed by what data would be useful for the expert. For example, if state value approximations are required, the policy might be expressed implicitly through a $Q$ function, as this can accelerate lookup.

## 3.2   Distributed Expert Iteration

Because our tree search is orders of magnitude slower than the evaluations made during training of the neural network, EXIT spends the majority of run time creating datasets of expert moves. Creating these datasets is an embarassingly parallel task, and the plans made can be summarised by a vector measuring well under 1KB. This means that EXIT can be trivially parallelised across distributed architectures, even with very low bandwidth.

## 3.3   Online expert iteration

In each step of EXIT, Imitation Learning is restarted from scratch. This throws away our entire dataset. Since creating datasets is computationally intensive this can add substantially to algorithm run time.

The online version of EXIT mitigates this by aggregating all datasets generated so far at each iteration. In other words, instead of training $\hat{\pi}_i$ on $\mathcal{D}_i$, we train it on $\mathcal{D} = \cup_{j \leq i} \mathcal{D}_j$. Such dataset aggregation is similar to the DAGGER algorithm [5]. Indeed, removing the expert improvement step from online EXIT reduces it to DAGGER.

Dataset aggregation in online EXIT allows us to request fewer move choices from the expert at each iteration, while still maintaining a large dataset. By increasing the frequency at which improvements can be made, the apprentice in online EXIT can generalise the expert moves sooner, and hence the expert improves sooner also, which results in higher quality play appearing in the dataset.

# 4 Imitation Learning in the game Hex

We now describe the implementation of EXIT for the board game Hex. In this section, we develop the techniques for our Imitation Learning step, and test them for Imitation Learning of Monte Carlo Tree Search (MCTS). We use this test because our intended expert is a version of Neural-MCTS, which will be described in section 5.

## 4.1 Preliminaries

### Hex

Hex is a two-player connection-based game played on an $n \times n$ hexagonal grid. The players, denoted by colours black and white, alternate placing stones of their colour in empty cells. The black player wins if there is a sequence of adjacent black stones connecting the North edge of the board to the South edge. White wins if they achieve a sequence of adjacent white stones running from the West edge to the East edge. (See figure 1).

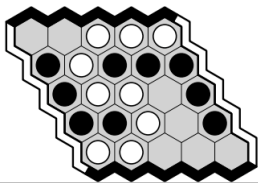

Figure 1: A $5 \times 5$ Hex game, won by white. Figure from Huang et al. [? ].

Hex requires complex strategy, making it challenging for deep Reinforcement Learning algorithms; its large action set and connection-based rules means it shares similar challenges for AI to Go. However, games can be simulated efficiently because the win condition is mutually exclusive (e.g. if black has a winning path, white cannot have one), its rules are simple, and permutations of move order are irrelevant to the outcome of a game. These properties make it an ideal test-bed for Reinforcement Learning. All our experiments are on a $9 \times 9$ board size.

### Monte Carlo Tree Search

Monte Carlo Tree Search (MCTS) is an any-time best-first tree-search algorithm. It uses repeated game simulations to estimate the value of states, and expands the tree further in more promising lines. When all simulations are complete, the most explored move is taken. It is used by the leading algorithms in the AAAI general game-playing competition [8]. As such, it is the best known algorithm for general game-playing without a long RL training procedure.

Each simulation consists of two parts. First, a *tree phase*, where the tree is traversed by taking actions according to a *tree policy*. Second, a *rollout phase*, where some *default policy* is followed until the simulation reaches a terminal game state. The result returned by this simulation can then be used to update estimates of the value of each node traversed in the tree during the first phase.

Each node of the search tree corresponds to a possible state $s$ in the game. The root node corresponds to the current state, its children correspond to the states resulting from a single move from the current state, etc. The edge from state $s_1$ to $s_2$ represents the action $a$ taken in $s_1$ to reach $s_2$, and is identified by the pair $(s_1, a)$.

At each node we store $n(s)$, the number of iterations in which the node has been visited so far. Each edge stores both $n(s,a)$, the number of times it has been traversed, and $r(s,a)$ the sum of all rewards obtained in simulations that passed through the edge. The tree policy depends on these statistics. The most commonly used tree policy is to act greedily with respect to the upper confidence bounds for trees formula [7]:

$$\text{UCT}(s,a) = \frac{r(s,a)}{n(s,a)} + c_b\sqrt{\frac{\log n(s)}{n(s,a)}} \tag{1}$$

When an action $a$ in a state $s_L$ is chosen that takes us to a position $s'$ not yet in the search tree, the rollout phase begins. In the absence of domain-specific information, the default policy used is simply to choose actions uniformly from those available.

To build up the search tree, when the simulation moves from the tree phase to the rollout phase, we perform an expansion, adding $s'$ to the tree as a child of $s_L$.[2] Once a rollout is complete, the reward signal is propagated through the tree (a *backup*), with each node and edge updating statistics for visit counts $n(s)$, $n(s,a)$ and total returns $r(s,a)$.

In this work, all MCTS agents use 10,000 simulations per move, unless stated otherwise. All use a uniform default policy. We also use RAVE. Full details are in the appendix. [9].

### 4.2 Imitation Learning from Monte Carlo Tree Search

In this section, we train a standard convolutional neural network[3] to imitate an MCTS expert. Guo et al. [11] used a similar set up on Atari games. However, their results showed that the performance of the learned neural network fell well short of the MCTS expert, even with a large dataset of 800,000 MCTS moves. Our methodology described here improves on this performance.

**Learning Targets**

In Guo et al. [11], the learning target used was simply the move chosen by MCTS. We refer to this as *chosen-action targets* (CAT), and optimise the Kullback–Leibler divergence between the output distribution of the network and this target. So the loss at position $s$ is given by the formula:

$$\mathcal{L}_{\text{CAT}} = -\log[\pi(a^*|s)]$$

where $a^* = \text{argmax}_a(n(s,a))$ is the move selected by MCTS.

We propose an alternative target, which we call *tree-policy targets* (TPT). The tree policy target is the average tree policy of the MCTS at the root. In other words, we try to match the network output to the distribution over actions given by $n(s,a)/n(s)$ where $s$ is the position we are scoring (so $n(s) = 10,000$ in our experiments). This gives the loss:

$$\mathcal{L}_{\text{TPT}} = -\sum_a \frac{n(s,a)}{n(s)}\log[\pi(a|s)]$$

Unlike CAT, TPT is cost-sensitive: when MCTS is less certain between two moves (because they are of similar strength), TPT penalises misclassifications less severely. Cost-sensitivity is a desirable property for an imitation learning target, as it induces the IL agent to trade off accuracy on less important decisions for greater accuracy on critical decisions.

In EXIT, there is additional motivation for such cost-sensitive targets, as our networks will be used to bias future searches. Accurate evaluations of the relative strength of actions never made by the current expert are still important, since future experts will use the evaluations of all available moves to guide their search.

**Sampling the position set**

Correlations between the states in our dataset may reduce the effective dataset size, harming learning. Therefore, we construct all our datasets to consist of uncorrelated positions sampled using an exploration policy. To do this, we play multiple games with an exploration policy, and select a single state from each game, as in Silver et al. [12]. For the initial dataset, the exploration policy is MCTS, with the number of iterations reduced to 1,000 to reduce computation time and encourage a wider distribution of positions.

We then follow the DAGGER procedure, expanding our dataset by using the most recent apprentice policy to sample 100,000 more positions, again sampling one position per game to ensure that there were no correlations in the dataset. This has two advantages over sampling more positions in the same way: firstly, selecting positions with the apprentice is faster, and secondly, doing so results in positions closer to the distribution that the apprentice network visits at test time.

### 4.3 Results of Imitation Learning

Based on our initial dataset of 100,000 MCTS moves, CAT and TPT have similar performance in the task of predicting the move selected by MCTS, with average top-1 prediction errors of $47.0\%$ and $47.7\%$, and top-3 prediction errors of $65.4\%$ and $65.7\%$, respectively.

However, despite the very similar prediction errors, the TPT network is $50 \pm 13$ Elo stronger than the CAT network, suggesting that the cost-awareness of TPT indeed gives a performance improvement. [4]

We continued training the TPT network with the DAGGER algorithm, iteratively creating 3 more batches of 100,000 moves. This additional data resulted in an improvement of 120 Elo over the first TPT network. Our final DAGGER TPT network achieved similar performance to the MCTS it was trained to emulate, winning just over half of games played between them ($87/162$).

## 5 Expert Improvement in Hex

We now have an Imitation Learning procedure that can train a strong apprentice network from MCTS. In this section, we describe our Neural-MCTS (N-MCTS) algorithms, which use such apprentice networks to improve search quality.

### 5.1 Using the Policy Network

Because the apprentice network has effectively generalised our policy, it gives us fast evaluations of action plausibility at the start of search. As search progresses, we discover improvements on this apprentice policy, just as human players can correct inaccurate intuitions through lookahead.

We use our neural network policy to bias the MCTS tree policy towards moves we believe to be stronger. When a node is expanded, we evaluate the apprentice policy $\hat{\pi}$ at that state, and store it. We modify the UCT formula by adding a bonus proportional to $\hat{\pi}(a|s)$:

$$\text{UCT}_{\text{P−NN}}(s, a) = \text{UCT}(s, a) + w_a \frac{\hat{\pi}(a|s)}{n(s, a) + 1}$$

Where $w_a$ weights the neural network against the simulations. This formula is adapted from one found in Gelly & Silver [9]. Tuning of hyperparameters found that $w_a = 100$ was a good choice for this parameter, which is close to the average number of simulations per action at the root when using 10,000 iterations in the MCTS. Since this policy was trained using 10,000 iterations too, we would expect that the optimal weight should be close to this average.

The TPT network's final layer uses a softmax output. Because there is no reason to suppose that the optimal bonus in the UCT formula should be linear in the TPT policy probability, we view the temperature of the TPT network's output layer as a hyperparameter for the N-MCTS and tune it to maximise the performance of the N-MCTS.

When using the strongest TPT network from section 4, N-MCTS using a policy network significantly outperforms our baseline MCTS, winning 97% of games. The neural network evaluations cause a two times slowdown in search. For comparison, a doubling of the number of iterations of the vanilla MCTS results in a win rate of 56%.

## 5.2 Using a Value Network

Strong value networks have been shown to be able to substantially improve the performance of MCTS [12]. Whereas a policy network allows us to narrow the search, value networks act to reduce the required search depth compared to using inaccurate rollout-based value estimation.

However, our imitation learning procedure only learns a policy, not a value function. Monte Carlo estimates of $V^{\pi^*}(s)$ could be used to train a value function, but to train a value function without severe overfitting requires more than $10^5$ independent samples. Playing this many expert games is well beyond our computation resources, so instead we approximate $V^{\pi^*}(s)$ with the value function of the apprentice, $V^{\hat{\pi}}(s)$, for which Monte Carlo estimates are cheap to produce.

To train the value network, we use a KL loss between $V(s)$ and the sampled (binary) result $z$:
$$\mathcal{L}_V = -z \log[V(s)] - (1 - z) \log[1 - V(s)]$$

To accelerate the tree search and regularise value prediction, we used a multitask network with separate output heads for the apprentice policy and value prediction, and sum the losses $\mathcal{L}_V$ and $\mathcal{L}_{\mathrm{TPT}}$.

To use such a value network in the expert, whenever a leaf $s_L$ is expanded, we estimate $V(s)$. This is backed up through the tree to the root in the same way as rollout results are: each edge stores the average of all evaluations made in simulations passing through it. In the tree policy, the value is estimated as a weighted average of the network estimate and the rollout estimate.[5]

# 6 Experiments

## 6.1 Comparison of Batch and Online ExIt to REINFORCE

We compare ExIt to the policy gradient algorithm found in Silver et al. [12], which achieved state-of-the-art performance for a neural network player in the related board game Go. In Silver et al. [12], the algorithm was initialised by a network trained to predict human expert moves from a corpus of 30 million positions, and then REINFORCE [3] was used. We initialise with the best network from section 4. Such a scheme, Imitation Learning initialisation followed by Reinforcement Learning improvement, is a common approach when known experts are not sufficiently strong.

In our batch ExIt, we perform 3 training iterations, each time creating a dataset of 243,000 moves.

In online ExIt, as the dataset grows, the supervised learning step takes longer, and in a naïve implementation would come to dominate run-time. We test two forms of online ExIt that avoid this. In the first, we create 24,300 moves each iteration, and train on a buffer of the most recent 243,000 expert moves. In the second, we use all our data in training, and expand the size of the dataset by 10% each iteration.

For this experiment we did not use any value networks, so that network architectures between the policy gradient and ExIt are identical. All policy networks are warm-started to the best network from section 4.

As can be seen in figure 2, compared to REINFORCE, ExIt learns stronger policies faster. ExIt also shows no sign of instability: the policy improves consistently each iteration and there is little variation in the performance between each training run. Separating the tree search from the generalisation has ensured that plans don't overfit to a current opponent, because the tree search considers multiple possible responses to the moves it recommends.

Online expert iteration substantially outperforms the batch mode, as expected. Compared to the 'buffer' version, the 'exponential dataset' version appears to be marginally stronger, suggesting that retaining a larger dataset is useful.

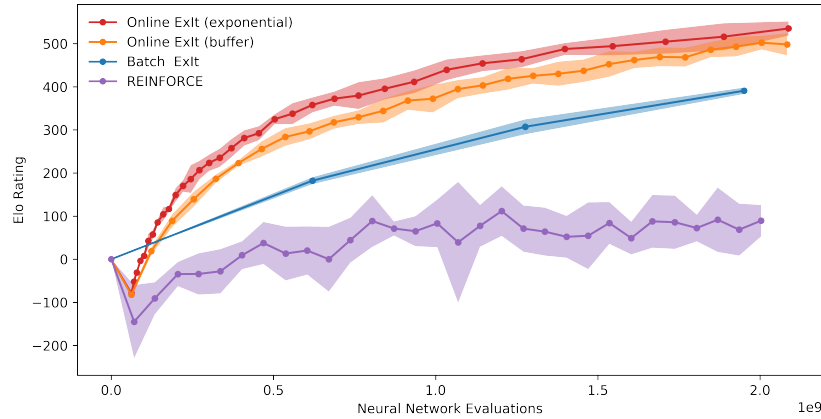

Figure 2: Elo ratings of policy gradient network and EXIT networks through training. Values are the average of 5 training runs, shaded areas represent $90\%$ confidence intervals. Time is measured by number of neural network evaluations made. Elo calculated with BayesElo [13]
.

## 6.2 Comparison of Value and Policy EXIT

With sufficiently large datasets, a value network can be learnt to improve the expert further, as discussed in section 5.2. We ran asynchronous distributed online EXIT using only a policy network until our datasets contained $\sim 550,000$ positions. We then used our most recent apprentice to add a Monte Carlo value estimate from each of the positions in our dataset, and trained a combined policy and value apprentice, giving a substantial improvement in the quality of expert play.

We then ran EXIT with a combined value-and-policy network, creating another $\sim 7,400,000$ move choices. For comparison, we continued the training run without using value estimation for equal time. Our results are shown in figure 3, which shows that value-and-policy-EXIT significantly outperforms policy-only-EXIT. In particular, the improved plans from the better expert quickly manifest in a stronger apprentice.

We can also clearly see the importance of expert improvement, with later apprentices comfortably outperforming experts from earlier in training.

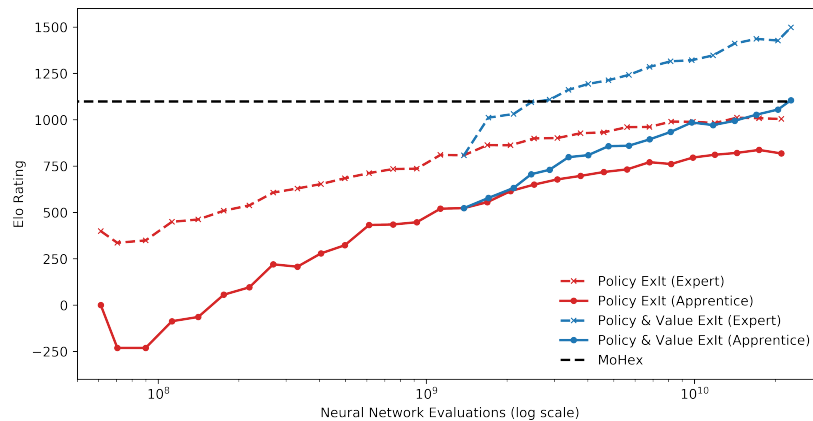

Figure 3: Apprentices and experts in distributed online EXIT, with and without neural network value estimation. MOHEX's rating (10,000 iterations per move) is shown by the black dashed line.

## 6.3 Performance Against MOHEX

Versions of MOHEX have won every Computer Games Olympiad Hex tournament since 2009. MOHEX is a highly optimised algorithm, utilising a complex, hand-made theorem-proving algorithm

which calculates provably suboptimal moves, to be pruned from search, and an improved rollout policy; it also optionally uses a specialised end-game solver, particularly powerful for small board sizes. In contrast, our algorithm learns tabula rasa, without game-specific knowledge beside the rules of the game. Here we compare to the most recent available version, MOHEX 1.0 [14].

To fairly compare MOHEX to our experts with equal wall-clock times is difficult, as the relative speeds of the algorithms are hardware dependent: MOHEX's theorem prover makes heavy use of the CPU, whereas for our experts, the GPU is the bottleneck. On our machine MOHEX is approximately 50% faster.[6] We tested EXIT against 10,000 iteration-MOHEX on default settings, 100,000 iteration-MOHEX, and against 4 second per move-MOHEX (with parallel solver switched on). EXIT won all matches using just 10,000 iterations per move, results in the table below:

| EXIT Setting | Time/move | EXIT win rate | MOHEX Setting | Solver | Time/move |
|---|---|---|---|---|---|
| $10^4$ iterations | $\sim 0.3$s | 75.3% | $10^4$ iterations | No | $\sim 0.2s$ |
| $10^4$ iterations | $\sim 0.3$s | 59.3% | $10^5$ iterations | No | $\sim 2s$ |
| $10^4$ iterations | $\sim 0.3$s | 55.6% | $4s$/move | Yes | $4s$ |

## 7    Related work

EXIT has several connections to existing RL algorithms, resulting from different choices of expert class. For example, we can recover a version of Policy Iteration [15] by using Monte Carlo Search as our expert; in this case it is easy to see that Monte Carlo Tree Search gives stronger plans than Monte Carlo Search.

Previous works have also attempted to achieve Imitation Learning that outperforms the original expert. Silver et al. [12] use Imitation Learning followed by Reinforcement Learning. Kai-Wei, et al. [16] use Monte Carlo estimates to calculate $Q^*(s, a)$, and train an apprentice $\pi$ to maximise $\sum_a \pi(a|s)Q^*(s, a)$. At each iteration after the first, the rollout policy is changed to a mixture of the most recent apprentice and the original expert. This too can be seen as blending an RL algorithm with Imitation Learning: it combines Policy Iteration and Imitation Learning.

Neither of these approaches is able to improve the original expert policy. They are useful when strong experts exist, but only at the beginning of training. In contrast, because EXIT creates stronger experts for itself, it is able to use experts throughout the training process.

AlphaGo Zero (AG0)[17], presents an independently developed version of ExIt, [7] and showed that it achieves state-of-the-art performance in Go. We include a detailed comparison of these closely related works in the appendix.

Unlike standard Imitation Learning methods, EXIT can be applied to the Reinforcement Learning problem: it makes no assumptions about the existence of a satisfactory expert. EXIT can be applied with no domain specific heuristics available, as we demonstrate in our experiment, where we used a general purpose search algorithm as our expert class.

## 8    Conclusion

We have introduced a new Reinforcement Learning algorithm, Expert Iteration, motivated by the dual process theory of human thought. EXIT decomposes the Reinforcement Learning problem by separating the problems of generalisation and planning. Planning is performed on a case-by-case basis, and only once MCTS has found a significantly stronger plan is the resultant policy generalised. This allows for long-term planning, and results in faster learning and state-of-the-art final performance, particularly for challenging problems.

We show that this algorithm significantly outperforms a variant of the REINFORCE algorithm in learning to play the board game Hex. The resultant tree search algorithm beats MoHex 1.0, indicating competitiveness with state-of-the-art heuristic search methods, despite being trained tabula rasa.

## Acknowledgements

This work was supported by the Alan Turing Institute under the EPSRC grant EP/N510129/1 and by AWS Cloud Credits for Research. We thank Andrew Clarke for help with efficiently parallelising the generation of datasets, Alex Botev for assistance implementing the CNN, and Ryan Hayward for providing a tool to draw Hex positions.

## Footnotes

[1]This reward may be decomposed as a sum of intermediate rewards (i.e. $R = \sum_{t=0}^{T} r_t$)

[2]Sometimes multiple nodes are added to the tree per iteration, adding children to $s'$ also. Conversely, sometimes an *expansion threshold* is used, so $s_L$ is only expanded after multiple visits.

[3]Our network architecture is described in the appendix. We use Adam [10] as our optimiser.

[4]When testing network performance, we greedily select the most likely move, because CAT and TPT may otherwise induce different temperatures in the trained networks' policies.

[5]This is the same as the method used in Silver et al. [12]

[6]This machine has an Intel Xeon E5-1620 and nVidia Titan X (Maxwell), our tree search takes 0.3 seconds for 10,000 iterations, while MOHEX takes 0.2 seconds for 10,000 iterations, with multithreading.

[7]Our original version, with only policy networks, was published before AG0 was published, but after its submission. Our value networks were developed before AG0 was published, and published after Silver et al.[17]

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
