[Supplementary Material]

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

# A    Comparison to AlphaGo Zero

Silver et al. [17] independently developed the EXIT algorithm, and applied it to Go. The result, AlphaGo Zero, outperformed the previous state-of-the art AlphaGo program while learning tabula rasa. Here we enumerate some of the similarities and differences between our implementations of the algorithm.

Both algorithms use tree policy targets for training the apprentice policy, however AlphaGo Zero uses Monte Carlo value estimates from the expert for the value network's training target, we use estimates from the apprentice. With approximately 100,000 times more computation during training than we used, the extra cost of such estimates is presumably outweighed by their greater accuracy. AlphaGo Zero includes all positions from all games in its dataset, whereas our positions are chosen to be independent samples from the exploration distribution.

In our implementation, we use the KL loss for both value and policy losses; AlphaGo Zero uses mean-square error for the value prediction. AlphaGo Zero also uses L2 regularisation of their network parameters, we use early stopping instead, and reinitialise neural network weights at each iteration before retraining on new data.

AlphaGo Zero uses a residual neural network with 79 layers, and significantly more units in each layer. Compared to a more standard CNN such as ours, they find significantly improved IL performance.

Our MCTS makes use of RAVE and rollouts, whereas AlphaGo Zero relies entirely on its value networks for evaluating moves. We also make use of warm-starting, progressing from a vanilla-MCTS expert to using only a policy network, and finally to using both policy and value networks. These warm starts are not essential for learning to play Hex, but they save substantially on computation early in training.

In AlphaGo Zero, before each expert improvement, they include verification that the new expert is indeed superior to the previous. They only change to a new expert if it defeats the current expert. Such tests would be prohibitively expensive compared to our training times, and we didn't find them to be necessary.

Unlike AlphaGo Zero, during dataset creation, we perform MCTS synchronously with no speed reduction, via the techniques described in appendix B.

When creating training data, AlphaGo Zero adds Dirichlet noise to the prior policy at the root node of the search. This guarantees that all moves may be tried by the tree search, thus avoiding local optima. We do not use this trick, and our UCT formula includes a more usual exploration term, which guarantees that every move at the root will be attempted at least once, and so the apprentice policy will never completely disregard a move. We do not know whether noise at the root would help in Hex.

# B    Fast calculation of expert moves

Because calculation of neural networks is faster when done in batches, and is performed on a GPU, most implementations of N-MCTS calculate their neural networks asynchronously: when a node is expanded, the position is added to a GPU calculation queue, but search continues. Once the queue length reaches the desired batch size $B$, the neural network policy can be calculated for the first $B$ states on the queue, and the information is added to the appropriate nodes.

Compared to waiting until the evaluation has taken place, this asynchronous neural net calculation substantially increases the rate at which MCTS iterations can take place: batching neural net calculations improves GPU throughput, and the CPU never sits idle waiting for evaluations. However, because search continues before evaluations are returned, suboptimal moves are made in the tree where prior information has not yet been calculated.

In the EXIT setting, we can avoid asynchronous N-MCTS, CPU idle time and small calculation batches on our GPU. This is because we are creating a large dataset of N-MCTS moves, and can calculate multiple moves simultaneously. Suppose we have a set $P$ of positions to search from, and that $|P| > 2B$. Each CPU thread gets a position $p_1$ from $P$, and continues search from that position until a NN evaluation is needed. It then saves the current search state (which can be expressed as a

single pointer to the current tree node), and submits the necessary calculation to the GPU queue. It then moves on to another position $p_2$ from $P$, which isn't awaiting a neural net evaluation.

## C  Monte Carlo Tree Search Parameters and Rapid Action Value Estimation (RAVE)

RAVE is a technique for providing estimates of the values of moves in the search tree more rapidly in the early stages of exploring a state than is achieved with UCT alone. This is important, because the Monte Carlo value heuristic requires multiple samples to achieve a low variance estimate of the value, which is particularly problematic when there are many actions available.

A common property of many games is that a move that is strong at time $t_2$ is likely to have also been strong at time $t_1 < t_2$. For instance, in stone placing games such as Go and Hex, if claiming a cell is useful, it may also have been advantageous to claim it earlier. RAVE attempts to exploit this heuristic to harness estimates for many actions from a single rollout.

RAVE statistics $n_{\text{RAVE}}(s)$, $n_{\text{RAVE}}(s, a)$ and $r_{\text{RAVE}}(s, a)$ are stored that correspond to the statistics used in normal UCT. After a simulation $s_1, a_1, s_2, a_2, ..., s_T$, with result $R$, RAVE statistics are updated as follows:

$$n_{\text{RAVE}}(s_{t_i}, a_{t_j}) := n_{\text{RAVE}}(s_{t_i}, a_{t_j}) + 1 \quad \forall \, t_i < t_j$$
$$r_{\text{RAVE}}(s_{t_i}, a_{t_j}) := r_{\text{RAVE}}(s_{t_i}, a_{t_j}) + R \quad \forall \, t_i < t_j$$
$$n_{\text{RAVE}}(s_{t_i}) := \sum_a n_{\text{RAVE}}(s_{t_i}, a) \quad \forall \, t_i$$

In other words, the statistics for state $s_{t_i}$ are updated for each action that came subsequently as if the action were taken first. This is also known as the *all-moves-as-first* heuristic, and is applicable in any domain where actions can often be transposed.

To use the statistics a $\text{UCT}_{\text{RAVE}}$ is calculated, and averaged with the standard UCT into the tree policy to give $\text{UCT}_{\text{U,RAVE}}$, which then chooses the action. Specifically:

$$\text{UCT}_{\text{RAVE}}(s, a) = \frac{r_{\text{RAVE}}(s, a)}{n_{\text{RAVE}}(s, a)} + c_b \sqrt{\frac{\log n_{\text{RAVE}}(s)}{n_{\text{RAVE}}(s, a)}}$$
$$\beta(s, a) = \sqrt{\frac{c_{\text{RAVE}}}{3n(s) + c_{\text{RAVE}}}}$$
$$\text{UCT}_{\text{U,RAVE}} = \beta(s, a)\text{UCT}_{\text{RAVE}}(s, a)$$
$$+ (1 - \beta(s, a))\text{UCT}(s, a)$$

The weight factor $\beta(s, a)$ trades the low variance values given by RAVE with the bias of that estimate. As the number of normal samples $n(s)$ increases, the weight given to the RAVE samples tends to 0. $c_{\text{RAVE}}$ governs how quickly the RAVE values are down-weighted as the number of samples increases.

When using a policy network, the formulae are:

$$\text{UCT}_{\text{P}-\text{NN}}(s, a) = \text{UCT}(s, a) + w_a \frac{\hat{\pi}(a|s, \tau)}{n(s, a) + 1}$$
$$UCT_{\text{P}-\text{NN,RAVE}}(s, a) = \beta(s, a)\text{UCT}_{\text{RAVE}}(s, a)$$
$$+ (1 - \beta(s, a))\text{UCT}(s, a)$$
$$+ w_a \frac{\hat{\pi}(a|s, \tau)}{n(s, a) + 1}$$

When using both policy and value estimates, the formulae are:

$$\mathrm{UCT}_{\mathrm{PV-NN}}(s,a) = \mathrm{UCT}(s,a) + w_a \frac{\hat{\pi}(a|s,\tau)}{n(s,a)+1} + w_v \hat{Q}(s,a)$$

$$UCT_{\mathrm{PV-NN,RAVE}}(s,a) = \beta(s,a)\mathrm{UCT}_{\mathrm{RAVE}}(s,a)$$
$$+ (1-\beta(s,a))\mathrm{UCT}(s,a)$$
$$+ w_a \frac{\hat{\pi}(a|s,\tau)}{n(s,a)+1} + w_v \hat{Q}(s,a)$$

Where $\hat{Q}(s,a)$ is the backed up average of the network value estimates at the edge $s,a$.

Table 1: Monte Carlo Tree Search Parameters. Vanilla-MCTS refers to the parameters used in section 4. N-MCTS parameters are for when only a policy network is used and when both policy and value networks are used.

| Parameter | Vanilla-MCTS | N-MCTS (policy) | N-MCTS (policy & value) |
|---|---|---|---|
| Iterations | 10,000 | 10,000 | 10,000 |
| Exploration Constant $c_b$ | 0.25 | 0.05 | 0.05 |
| $c_{\mathrm{RAVE}}$ | 3000 | 3000 | 3000 |
| Expansion Threshold | 0 | 1 | 1 |
| NN Weight $w_a$ | N/A | 100 | 100 |
| NN Output Softmax Temperature $\tau$ | N/A | 0.1 | 0.1 |
| Value Network Weight $w_v$ | N/A | N/A | 0.75 |

## D   Neural Network Architecture

**Input Features**. We use the same state representation as Young et al. [18]: a two-dimensional state of $9 \times 9$ Hex board is extended to a 6 channel input. The 6 channels represent: black stone locations, white stone locations, black stones connected to the north edge, black stones connected to the south edge, white stones connected to the west edge and white stones connected to the east edge.

In line with Young et al. [18], to help with the use of convolutions up to the board edge, we also expand the board, adding two extra rows or columns to each side. On the extra cells thus created, we add dummy stones: along the North and South edges, black stones, along the East and West edges, White stones. In each corner of the padding, we 'place both a black and a white stone'. The resultant encoding of the board is shown in figure 4.

Playing Hex on this expanded board, with the dummy stones providing connections in the same way as stones played by players, does not change the game, but it means that convolutions centred at the edge of the board have more meaningful input than would be provided by zero-padding these cells.

**Neural network architecture**. Our network has 13 convolution layers followed by 2 parallel fully connected softmax output layers.

The parallel softmax outputs represent the move probabilities if it is black to move, and the move probabilities if it is white to move. Before applying the softmax function, a mask is used to remove those moves which are invalid in the current position (i.e. those cells that already have a stone in them).

When also predicting state value, we add two more outputs, each outputting a scalar with a sigmoid non-linearity. This estimates the winning probability, depending on which player is next to move.

Because the board topology is a hexagonal grid, we use hexagonal filters for our convolutional layers. A $3 \times 3$ hexagonal filter centred on a cell covers that cell, and the 6 adjacent cells.

In convolution layers 1-8 and layer 12, the layer input is first zero padded and then convolved with 64 $3 \times 3$ hexagonal filters with stride 1. Thus the shape of the layer's output is the same as its input. Layers 9 and 10 do not pad their input, and layers 11 and 13 do not pad, and have $1 \times 1$ filters.

Figure 4: The 6-channel encoding used as input for our NN

Our convolution layers use Exponential linear unit (ELU) [21] nonlinearities. Different biases are used in each position for all convolution layers and normalisation propagation [20] is applied in layers 1-12.

This architecture is illustrated in figure 5.

**Training details**. At each training step, a randomly selected mini batch of 250 samples is selected from the training data set and Adam [10] is used as optimiser. We regularise the network with early stopping. The early stopping point is the first epoch after which the validation errors increase 3 times consecutively. An epoch is one iteration over each data point in the data set.

6 channels representation

convolution scan

Convolution Layer 1-12

NP activation

non-linear function

normalisation propagation

x12 layers

Convolution Layer 13

if Black needs to move

if White needs to move

Fully Connected Layer

Fully Connected Layer

Fully Connected Layer

Fully Connected Layer

Policy for Black

Value for Black

Policy for White

Value for White

Figure 5: The NN architecture

# E  Matches between EXIT and MOHEX

EXIT is clearly stronger than MOHEX 1.0. MOHEX 2.0 is ∼250 Elo stronger than MOHEX 1.0 on $11 \times 11$, with the strength difference usually slightly lower on smaller boards. Although the most recent versions of MOHEX are not available for comparison, we conclude that EXIT is competitive with state of the art for (scalable) heuristic search methods, but cannot say whether it exceeds it, particularly for larger 11x11 or 13x13 boards, which are more typical for tournament play. Future work should verify the scalability of the method, and compare directly to state-of-the-art methods.

We present six games between EXIT (10,000 iterations per move) and MOHEX[8] (100,000 iterations per move), from a match of 162 games (consisting of one game each as black per legal opening move), of which EXIT won 59.3%. Here MOHEX only uses its MCTS; in tournament play it uses DFPN solver in parallel to the MCTS. The games were chosen with 'even' opening moves that don't give either player a large advantage, and to show some of the relative strengths and weaknesses of both algorithms.

Figure 6: EXIT (black) vs MOHEX (white)

Figure 7: MoHex (black) vs ExIt (white)

Figure 8: ExIt (black) vs MoHex (white)

Figure 9: MOHEX (black) vs EXIT (white)

Figure 10: EXIT (black) vs MOHEX (white)

Figure 11: MoHex (black) vs ExIt (white)