[Reviews · NeurIPS 2017]

Reviewer 1



SUMMARY: The paper proposes an algorithm that combines imitation learning with tree search, which results in an apprentice learning from an ever-improving expert. A DQN is trained to learn a policy derived from an MCTS agent, with the DQN providing generalisation to unseen states. It is also then used as feedback to improve the expert, which can then be used to retrain the DQN, and so on. The paper makes two contributions: (1) a new target for imitation learning, which is empirically shown to outperform a previously-suggested target and which results in the apprentice learning a policy of equal strength to the expert. (2) using the apprentice to improve the expert after each imitation learning step is shown to result in strong play against a policy gradient algorithm for the game Hex. COMMENTS: I found the paper generally well-written, clear and easy to follow, barring Section 6. The idea of combining reactive-style approaches with explicit planning techniques is an interesting one, and makes sense intuitively (I particularly liked the link to the different systems of human thinking), although the downside here is that we then require a model of the environment, which negates the benefits of using model-free approaches (like DQN) in the first place. A few questions that I had about the paper: 1. On line 69, some of the drawbacks of RL in general are stated. What guarantees, though, are there that these issues don't apply to ExIt as well? 2. On line 212, why is it advantageous or desirable for the positions in the training set to be similar to those in the test set? 3. I found section 6 lacking in detail, especially for a purely empirical paper. In particular, was the version of ExIt presented here the online version? On lines 122-125, some reasons are provided as to why an online variant may be superior, but there is no experiment validating this. I would therefore expect to see maybe an online version, and a version where the data is discarded after each iteration in the results graph, which could then show that ExIt is better than policy gradients, as well as the advantage (or not) of the online version. 4. I do not follow the results in section 6.2. What were the "reference agents" mentioned in the caption? Which network is being referred to on line 267? How many games were played? What were the standard errors on those results? What is "NN evaluations"? Some remarks on the actual writing: 1. For an arbitrary policy, the definition of the advantage function appears incorrect on line 50. It should be defined as A^\pi(s, a) = Q^\pi(s,a ) - V^\pi(s), where V\pi(s) = Q^\pi(s,\pi(s)) which is not necessarily equal to the max of one-step lookahead as is currently written. 2. In the pseudocode on line 80, there is a function called "Targets". Since it is only made clear two pages later what "Targets" actually refers to, it may be better to briefly mention its meaning here. 3. The exploration constant c_b in the UCB1 formula on line 162 is not explained. Also, was this set to zero because RAVE was used? 4. On line 232, should it not read "When a node is TRAVERSED"? The word "expanded" gives the impression that it is in the expansion phase of MCTS where a child is randomly added to the tree, as opposed to the tree phase where UCB1 selects the next child. There were also a few minor typographical errors: 1. There are missing full stops on lines 101 and 171 2. The footnote on line 167 should appear after the full stop 3. The semicolon on line 187 should be a comma 4. Line 254: "by network trained" => "by a network trained" 5. In the footnote on line 176, "ADAM" has been capitalised, but in the supplementary material it is referred to as "adam" 6. The reference on line 340 has the first names of the authors listed, instead of their surnames.

Reviewer 2



It's not clear to me what the major distinction is between the proposed approach and AlphaGo. It seems to me that "expert iteration" is fundamentally also a variation of using Monte Carlo Tree Search as was in the case of AlphaGo

Reviewer 3



In general, I think it is a good paper. I like the idea, and find the paper well written. It would be nice to see how well it does in practice. Why not share a few games played by raw nets and N-MCTS ? What is SOTA in computer Hex ? How is this doing compared to SOTA ? Why Hex ? Why not Go, as this arguably an improvement of AlphaGo N-MCTS ? Also would like to see winrates for final N-MCTS + networks rather than just raw networks winrate. With more solid experiments / analysis it would have been an excellent paper and I would love to give it a higher rating. Also, can you explain how you convert the policy / softmax of TPT net into an advantage ? Are they just equal ? l 241. Ho I see, you mean the advantage is the logits of your TPT net ? Would be nice to make that explicit before. Also, this is a bit abusive. Why should the logits of your policy match the advantage of a Q value of that policy ? Would be cleaner to just not use that advantage terminology here and just stick to policy IMHO. l 245. Do you mean N-MCTS wins 97% against MCTS with the same number of simulations ? But presumably, N-MCTS is much slower to do one iteration, because of neural network evaluation. A more interesting datapoint here would be your N-MCTS winrate against your MCTS with fixed time per move. l 267. Same comment here. Are these head to head network games played with softmax or argmax ? Softmax is not really fair, it could be the two learning rules results in different temperatures. Figure 2. Why does your ExIt win curves consist in plateaus ? Would be nice to see the learning dynamics between each dataset iterations. What are the 'reference agents' ? Figure 2 shows that ExIt imrpove raw networks playning strength. Does this also actually improves the expert N-MCTS ? Would be interesting to see the winning rate curve for the N-MCTS experts. How much stronger is N-MCTS + ExIt network versus N-MCTS + IL network versus N-MCTS + IL->REINFORCE network ?